# Experimental and Numerical Study on Lightweight-Foamed-Concrete-Filled Widened Embankment of High-Speed Railway

**DOI:** 10.3390/ma17184642

**Published:** 2024-09-21

**Authors:** Didi Hao, Changqing Miao, Shisheng Fang, Xudong Wang, Qiaoqiao Shu

**Affiliations:** 1Key Laboratory of Concrete and Prestressed Concrete Structures of Ministry of Education, Southeast University, Nanjing 211189, China; hdd@seu.edu.cn (D.H.); civilwang@seu.edu.cn (X.W.); 2School of Civil Engineering, Southeast University, Nanjing 211189, China; 3School of Civil and Hydraulic Engineering, Hefei University of Technology, Hefei 230009, China; shisheng_fang@163.com; 4Brother Andrew Gonzalez FSC College of Education, De La Salle University, Manila 0920, Philippines; qiaoqiao_shu@dlsu.edu.ph

**Keywords:** lightweight foamed concrete, widened embankment, high-speed railway, performance test, numerical modeling, parametric study

## Abstract

To study the performance of lightweight foamed concrete (LWFC) in widened embankments of high-speed railways, this study first conducted numerous strength, permeability, and water immersion tests to investigate the mechanical properties and water resistance of LWFC with designed dry densities of 550, 600, and 650 kg/m^3^. Secondly, a field test was performed to analyze the behavior of the deformation and the internal pressure within the LWFC-filled portions. Furthermore, a parametric study via numerical modeling was performed to investigate the effects of four key factors on the performance of the LWFC-filled, widened embankments. Results showed that LWFC possesses adequate bearing capacity and impermeability to meet high-speed railway embankment widening requirements. However, water seepage reduces LWFC strength. The additional pressure from LWFC filling increases initially but then decreases once dehydration occurs. The settlement induced by LWFC accounted for 71% of the total filling height, which is only 37.5% of the total settlement after construction. The parametric study results show that the maximum settlement of widened and existing portions induced by LWFC was 46.3–49.6% and 48.3–53.2% of those induced by traditional fillers due to the LWFC’s lower density as well as their better self-supporting ability. Making an appropriate reduction in the thickness of the retain wall installed against the LWFC-filled widened embankment of the high-speed railway generates a few variations in the lateral deformation of the wall. Furthermore, the effects of the pile offset on the deformation of the LWFC-filled embankment were more sensitive compared to the diameter of the piles.

## 1. Introduction

The total length of operational railways in China has reached 150,000 km by March 2023, of which approximately 40,000 km are high-speed railways. In recent years, high-speed railways have been constructed at an accelerating rate to cater to the demands of rapid development in China. Single-track construction and embankment widening are currently the main measures adopted to expand the high-speed railway network, and embankment widening is widely applied in China due to the advantages of high efficiency and low costs. However, additional settlement occurs between the widened embankment and the existing embankment due to the non-symmetric loading generated by the self-weight of the widened portion of the embankment [1,2,3], as shown in Figure 1.

Numerous case studies [3,4], numerical modeling tests [5,6,7], and model tests [8,9,10] have been conducted to analyze the performance of widened embankments using various methods to control the additional settlement. Research has mainly focused on improving the vertical stiffness of the foundation under the widened embankment through structural reinforcement. In the case of studies, geogrid layers have been installed under the widened embankment to produce anchorage to reduce differential settlement between the existing embankment and the widened embankment [11,12]. CFG piles [13], PTC piles [14], and open-ended PHC piles [15] provide stiffness and shear resistance to the deformation and shear stress induced by widened embankments. Optimizing the column spacing is one effective means of improving the deformation between the existing and widened portions of the embankment during pile reinforcement [16]. Surcharge preloading on weak soil under the embankment remarkably reduces the differential settlement between the old embankment and the widened section by increasing the compression modulus and cohesion of the soil after consolidation [17]. This method has the same mechanisms as soil replacement, drainage consolidation, and soil compaction. Based on the aforementioned studies, different layouts and parametric of reinforcements, including piles and geogrid layers under the widened embankment, have been investigated using numerical modeling [16,17,18,19,20,21]. Appropriate adjustment of pile spacing, pile length, and geogrid structure type reduce the vertical and horizontal deformation induced by embankment widening, and the use of dry-jet mixing columns to enhance the foundation soil under the widened embankment is more effective in reducing the deformation than the use of prefabricated vertical drains. Field and model tests have been presented to research the deformation and distribution of force in widened embankments. In particular, field tests [6,15] have been conducted to study excess pore pressure, lateral soil deformation, and settlement under the widened embankment, and results have revealed that the effect of pile group installation under the widened embankment on the existing embankment can be ignored, the pavement upon the embankment must be pulled apart when the differential settlement reaches tens of centimeters, and geogrid reinforcement adopted between the existing and widened portions of the embankment can effectively alleviate the stress of the pavement. The use of geogrid-reinforced embankment [22], pipe–pile composite foundation [23], and EPS-block geofoam [24] for embankment widening reduces the settlement and shear force between the new portion and existing embankment, as verified using the centrifuge model tests.

Apart from the methods investigated above for reinforcing foundations under the widened embankment, new building materials with well-adapted performance to improve the geological properties and reduce unmanageable deformation have been studied by researchers. Lightweight foamed concrete (LWFC), a new type of material with lower density (Figure 2) and higher strength compared to traditional fillers, has been widely used in oilfield cementing, soft base reinforcement, and embankment widening [25]. Research on LWFC has mainly focused on performance experiments [26], performance strengthening by using admixture [27] and evaluating the effect [28] in engineering. Design specifications have been established for the application of LWFC on highways in China [29]. However, few studies [30,31] have been conducted on railway embankment widening projects. Learning from the successful application of LWFC in highway embankments, the use of LWFC in railway embankments has attracted increased research interest in recent years [31,32].

By now, studies on the application of LWFC are still primarily focused on highway subgrade filling, widening, and building material usage. With the continuous expansion and construction of high-speed railways, widening routes adjacent to existing embankments has gradually become a new trend. However, traditional embankment fillers are relatively heavy, requiring substantial costs to reinforce the widened portion to reduce the overall settlement caused by the additional load. Thus, it is urgent to find a lightweight, high-strength alternative filler to address the issue. Based on the idea that LWFC filler may be considered a promising material for embankment widening in high-speed railways, this work conducted extensive experimental studies, including the performance tests conducted to study the strength and water permeability of LWFC. In addition, a field test was performed to investigate the behavior of LWFC-filled widened embankments, for which a three-dimensional numerical modeling was established. Then, a parametric study was conducted to study the influence of four key factors on the performance of widened embankments.

## 2. Performance Test of LWFC

### 2.1. Preparation of the Specimen

The LWFC adopted in this study is composed of cement and foam mixed with water, of which ordinary portland cement of grade P·O42.5R was selected in accordance with literature [33]. The LWFC specimens adopted in this study were designed, implemented, and cured with reference to the literature [33] as well. Figure 3 illustrates the preparation procedure for the LWFC specimens in this study. The foam used was made by a physical foaming machine with a model of ZS-100A diluting foaming agent mixed with compressed air to produce uniform and continuous foam. The type of foaming agents were compound blowing agents provided by The Fourth Engineering Co., Ltd., of CTCE Group, located in in Hefei, China, and the physical properties of the foam produced from them are shown in Table 1. The mix proportions of LWFC for different design dry densities are presented in Table 2. The water-cement ratio was designed with a fixed value of 0.5 for all mixtures. The mixing of LWFC is completed using a CMP500 planetary mixer, with the ambient temperature controlled at 20 °C ± 2, and the mixing time controlled at 15 min. The specimens tested can be made only when the properties of cement slurry and foam meet the performance requirements and Table 1, respectively. Numerous cube samples (100 mm × 100 mm × 100 mm), rectangular samples (40 mm × 40 mm × 160 mm), as well as circular-truncated-cone samples (185 mm, 175 mm for the bottom, top diameter, and 150 mm high), were poured for strength tests, permeability tests, and water immersion tests. The specimens were maintained in accordance with the literature [33] and demolded after 14 days.

Based on the former literature [32], high stiffness, strength, and water resistance are required to meet the requirements for embankments of high-speed railways. Numerous tests for the compressive strength, tensile strength, permeability, and strength under water treatment were carried out to investigate the mechanical performance and anti-permeability of LWFC, which is supposed to be an ideal filler for embankment widening of high-speed railways with dry densities of 550, 600, and 650 kg/m^3^. The testing site is located in the Building Materials Laboratory at Hefei University of Technology.

To improve the reliability of the testing program, the loading machines were thoroughly inspected and calibrated before each test. The test was conducted only when the corresponding equipment was confirmed to be operating safely and normally. The specimens used in different tests were inspected for quality and appearance before loading, and defective specimens were discarded. During the testing process, two individuals are assigned to read and cross-check the instrument readings.

### 2.2. Compressive Strength Test

As shown in Figure 4, cubic specimens with a side length of 100 mm were tested under compression by using the TSZ triaxial apparatus to determine the material strength and elastic modulus [33,34]. A load capacity of 100 kN with an increased loading rate of 2 k/s was applied. Three specimen tests were carried out for each design dry density, wherein the average value was used as the representative result. The testing of three sets of specimens was completed in a short time under dry conditions at approximately 25 °C. During the test, the axial force meter and the adscititious displacement meter were used to test the strain data under a specific load so that the mechanical property parameters of the materials could be further obtained. The two instruments were inspected and calibrated before the test.

### 2.3. Tensile Strength Test

Generally, splitting tests, bending tests, and uniaxial direct tensile tests are widely applied to determine the tensile strength of the brittle material, including the LWFC [33,34]. However, local failure often occurs near the fixture in the direct tensile test for LWFC, thus resulting in inefficiency for test conduction. Hence, the splitting test and bending test in this study were conducted to determine the tensile strength of LWFC. The 100-mm-long specimens, in the splitting test, were split along the center of the cube by 20 mm wide rigid strips with a load capacity of 50 kN and an increased loading rate of 0.5 kN/s, as shown in Figure 5a. The testing conditions, including temperature and humidity, were set to be the same as those in the compressive strength test. The tensile strength of LWFC can be obtained as follows:(1)$$f_{ts}=\frac{2P}{\pi A},$$
where *f_ts_* is the splitting tensile strength (MPa), *P* is the ultimate failure load of the specimen (N), and *A* is the area of the split surface (1500 mm^2^).

In the bending test, specimens with a size of 40 mm × 40 mm × 160 mm supported by two steel rods with 120 mm apart were tested under a bending load on their mid-span with a load capacity of 10 kN and an increased loading rate of 50 N/s, as shown in Figure 5b. The tensile strength can be calculated as follows:(2)$$f_{f}=\frac{1.5PL}{BH^{2}},$$
where *f_f_* is the flexural tensile strength (MPa), *P* is the ultimate failure load of the specimen (N), *L* is the distance between the supports (mm), *B* and *H* are the width (mm) and height (mm) of the specimen, respectively.

### 2.4. Permeability Test

Specimens with a lower diameter of 185 mm, an upper diameter of 175 mm, and a height of 150 mm were tested under different hydraulic pressures using a permeability testing machine to study the impermeability of LWFC with dry density of 550, 600, and 650 kg/m^3^. The testing flow is shown in Figure 6. The LWFC specimen was first treated with sealing wax on the side area to prevent water from infiltrating laterally during the test. Then, the specimens were pressed into the mold and placed on the permeameter. In this test, water pressures of 20, 40, and 60 kPa, were applied to specimens to obtain the permeability coefficient, and each test was performed with five specimens. Finally, the average value of each test was selected as a result as well.

The permeability coefficient can be calculated from tests as follows:(3)$$K_{q}=\frac{QH\gamma_{\omega }}{PA},$$
where *K_q_* is the permeability coefficient (m/s), *Q* is the amount of water passing through the specimen per unit time (m^3^), *H* is the height of the specimen (m), *A* is the average cross-sectional area of the specimen (m^2^), *P* is the difference in the hydraulic pressure between the two ends of the specimen (kPa), and $\gamma_{\omega }$ is the bulk density of water (kN/m^3^).

### 2.5. Water Immersion Test

Embankments in high-speed railway corridors are often exposed to a low-pressure water environment. The water immersion tests were carried out to investigate the influence of the soaking environment on the compressive strength of LWFC, as shown in Figure 7. The specimens were immersed in shallow water, and their weights were measured every 2 h until the total weight change for ten consecutive measurements was less than 2%, and then the compression test was conducted. 

### 2.6. Analysis of Testing Results

#### 2.6.1. Compressive Strength and Tensile Strength of LWFC without the Effect of Water

As can be seen in Figure 8a, the failure of specimens under the compressive tests occurred with a major fracture vertically distributed along the central side of the block, localized crushing of which was observed especially in a lower density. The compressive strength and elastic modulus of LWFC increase linearly with increases in dry density. The mean values of compressive strength tested from LWFC specimens with dry densities of 550, 600, and 650 kg/m^2^ were about 1.0 MPa, 1.6 MPa, and 2.0 MPa, respectively. The values of elastic modulus, correspondingly, were close to 94.5 MPa, 104.9 MPa, and 122.8 MPa, respectively. The compressive strength as well as the elastic modulus of LWFC exhibited a good linear relationship with dry density, as illustrated in Figure 9 and Figure 10, respectively The empirical equations of compressive strength and elastic modulus with dry density of LWFC were derived through numerical fitting, as shown in Formulas (4) and (5), respectively.
(4)$$\dot{S}=9.4\cdot \rho \times 10^{-3}+o(\rho )$$
(5)$$\dot{E}=0.256\cdot \rho +o(\rho )$$
where $\dot{S}$ and $\dot{E}$ are the compressive strength (MPa) and elastic modulus (MPa), respectively, *ρ* is the dry density (kg/m^3^), and o(ρ) is the intercept with a tiny value.

During the splitting tests, specimens with different dry densities all failed with one vertical crack along the center of the loading points, and no cracking occurred in other areas. And partial spalling occurred on the surface of the specimens adjacent to the lower cushion, as shown in Figure 8b. Some ductility was observed in the failure process due to the bubbles distributed in the specimen, thus resulting in the transfer hysteresis of compressive stress. During the bending tests, the sudden brittle fracture at the bottom of the span, soon after the loading began, was recorded from tested specimens with different dry densities, as illustrated in Figure 8c. Both the splitting tensile strength and the bending tensile strength of LWFC exhibited a good linear relationship with dry density, as illustrated in Figure 11 and Figure 12, respectively. And the function expression between them and the dry density obtained by data fitting are as follows:(6)$$\dot{f_{ts}}=2.12\cdot \rho {\;\times \;10}^{-3}-0.92,$$
(7)$$\dot{f_{f}}=-3.73\cdot \rho {\;\times \;10}^{-3}-1.70$$
where $\dot{f_{ts}}$ and $\dot{f_{f}}$ are the splitting tensile strength (MPa) and bending tensile strength (MPa), respectively, and *ρ* is the dry density (kg/m^3^).

#### 2.6.2. Permeability Coefficient of LWFC

During the permeability tests, an osmotic phenomenon was observed between 11 and 15 min after the testing began. This phenomenon was captured in the marginal region located on the upper surface of the specimens, as depicted in Figure 13. The early detection of this osmotic activity indicated the initial stages of water movement through the samples, highlighting the dynamic changes occurring within the material’s structure. As the tests progressed further, the amount of water permeating through the samples per unit time gradually stabilized. Around 30 to 37 min after the tests commenced, a steady-state flow condition was reached. This steady-state condition is significant as it marks the point where the permeability rate remains consistent over time, suggesting that the internal structure of the samples has equilibrated under the testing conditions. To systematically analyze and quantify the impact of bubble distribution on permeability, an allometric distribution model was applied to fit the experimental data. This model provided a mathematical framework to describe the relationship between bubble distribution and permeability variations within the tested blocks. The fitting of the data using the allometric distribution model is illustrated in Figure 14, which shows the correlation between the observed data and the model’s predictions. The derived fitting equation from this model is presented below, encapsulating the relationship between the structural characteristics of the LWFC and its permeability properties. This model and its equation provide valuable insights into the material’s behavior, enabling a better understanding of how bubble distribution affects the permeability of LWFC. The variability in the time taken to reach this steady-state flow condition can be attributed to the heterogeneous distribution of bubbles within the Lightweight Foamed Concrete (LWFC). The presence of bubbles and their uneven distribution across different regions of the material lead to variations in permeability. Specifically, areas with a higher concentration of bubbles exhibited different permeability characteristics compared to regions with fewer bubbles. This spatial variation in bubble distribution within the LWFC is a critical factor influencing its overall permeability properties.
(8)$$\dot{K_{q}}=\left(1.04\cdot 10^{-3}\right)\cdot {\left(\rho \cdot 10^{-3}\right)}^{-14.4}$$
where $\dot{K_{q}}$ is the permeability coefficient (10^−3^ m/s), and *ρ* is the dry density of LWFC (kg/m^3^).

#### 2.6.3. Compressive Strength of LWFC with the Effect of Water

The seepage line of soaked LWFC specimens with stable weight in the immersion tests is shown in Figure 15a. Minor water absorption occurred on the surface of the specimens under a low-pressure water environment, thus indicating that LWFC provides sufficient impermeability to resist liquid infiltration and erosion during the service life of the high-speed railway. The failure of soaked specimens during the compressive strength test was accompanied by localized disintegration on the surface close to the loading area, as can be seen in Figure 15b. As shown in Table 3, the compressive strength recorded from specimens with dry density of 550 kg/m^3^, 600 kg/m^3^ and 650 kg/m^3^ using for water immersion tests fell by 11.7%, 8.8%, and 6.2% compared to those without the effect of water, respectively. 

## 3. Field Test

In the previous chapter, performance tests were performed to investigate the mechanical properties and water resistance of LWFC with various designed dry densities. However, it is not rigorous to judge the suitability of LWFC as a widening filler in high-speed railway based solely on laboratory-derived material performance data. Therefore, we need to conduct large-scale field tests to fully demonstrate the practical feasibility of LWFC. A field test was conducted to further investigate the performance of LWFC-filled, widened embankments adjacent to existing high-speed railway tracks. The engineering site selected for the field test is located in front of the Fei-dong Station China, where the embankment widening was applied using LWFC filler with a design dry density of 580 kg/m^3^ and 640 kg/m^3^ to merge with the existing embankment of Hefei-Nanjing high-speed railway. The existing portion of the embankment is filled with schist fillers, and the settlement has reached stability. 

### 3.1. Widened Embankments

As shown in Figure 16a, the widened embankment consists of a lower LWFC layer with a height of 4.1–6.3 m and an upper layer of A, B group fillers with a height of 2.6 m. The foundation is reinforced with a cement mixing pile with a diameter and offset of 0.5 and 1 m, respectively. A 0.6-m-layer geotechnical cushion consisting of geogrid and gravel is used between the reinforcement piles and the LWFC embankment. A 0.5 m thick retaining wall with 2 m long anchor bars embedded in LWFC is the outside of the widened embankment. The construction process of the LWFC-filled embankment is shown in Table 4.

### 3.2. Site Conditions

The geological characteristics of the soil beneath the foundation at the test site were analyzed by borehole sampling and summarized as follows: deep silty clay (K2c) at 3.8 m, deep soft clay (Q4al) at 4.3 m, and intermediary weathered sandstone (Q3al) at depths greater than 4.3 m. A brief soil investigation summary provides useful geotechnical context for the test site. The physical properties of the soil layers were examined using undisturbed soil tests, and the average values for the soil properties for each layer are summarized in Table 5. Additionally, the surface above the foundation is relatively flat, and the groundwater table ranges from 1.2 to 5 m below the ground surface. 

### 3.3. Monitoring Variables in Field Test

In field test, two displacement meters, eight earth pressure cells, and two inclinometers were placed in the profile located at the site with the maximum filling height of the widened portion to determine the deformation and pressure inside the widened embankment during the filling stage. The locations of the measuring instruments are shown in Figure 16. The displacement meters were placed at the bottom of the widened embankment, with one numbered S-1 at the centerline of that and the other one numbered S-2 at the side of the base adjacent to the existing embankment, to measure the foundation settlement. The pressure cells numbered P-1 to P-8 were placed at the base and LWFC to measure the vertical pressure on the foundation and the horizontal pressure on the retain wall within the LWFC-filled widened embankment, respectively. The P-3,4,5 and the P-6,7,8 were installed in the foundation, and the 3 m above the base, in which the P-4,7 were installed at the same position with S-1 and the P-3,6 and P-5,8 were placed at the sides of the base. The P-1 and P-2 were placed at the top and mid-height of the retaining wall, respectively. The two inclinometers were mounted on the outermost pile of either side along a horizontal direction, with one numbered NSP closed to the existing embankment and the other numbered FSP.

## 4. Numerical Modeling

### 4.1. Model Mesh and Boundary Condition

A three-dimensional model was performed to simulate the behavior of the LWFC-filled, widened embankment selected for the field test using the FLAC-3D software 6.0 based on the finite differential method. The boundary effect, according to references [35], can be ignored when the size of the model is approximately 5–6 times larger than the region to be focused. Therefore, the length, width, and height of the model were 3 m, 154 m, and 112 m, respectively, as shown in Figure 17. The structural mesh consisting of hexahedron elements, with a size of 0.1 m in the region of widened embankment and 0.2 to 0.8 m in other areas, was presented to improve the computational precision and save the computed time. The mesh was established utilizing built-in meshing technology in FLAC software, where all sub-portions were set as assemblies composed of hexahedral elements. In accordance with the literature [35], the mechanical boundary conditions were set as follows: X, Y, and Z directions of deformation of nodes located at the bottom of the numerical model were fixed, and normal fixed deformation was applied to nodes on the four sides.

### 4.2. Constitutive Relationships and Properties of the Material

The piles and the retain wall were modeled as linear-elastic materials. The LWFC, the geotechnical cushion, the A,B group filler, the weathered rock and the existing embankment were modeled as elastic–plastic materials using the Mohr–Coulomb criterion, as shown in Figure 18a, where the potential function was described with two functions in the criterion, namely *g_s_* and *g_t_*, used to define the shear plastic flow and tensile plastic flow, respectively. The function *g_s_* corresponds to a non-associated law and has the following form:(9)$$g_{s}=-\sigma_{1}-\sigma_{3}\frac{1+\sin\varphi_{f}}{1-\sin\varphi_{f}}$$

The function *g_t_* corresponds to an associated flow rule Figure 18b and can be expressed as follows:(10)$$g_{t}=\sigma_{3}-\sigma_{t}$$
where $\sigma_{1}$ and $\sigma_{3}$ ($\sigma_{1}$ > $\sigma_{3}$) are the components of the generalized stress vector for this model, *φ_f_* is the dilation angle, and $\sigma_{t}$ is the tensile strength.

And the following seven parameters need to be defined in the Mohr–Coulomb model: *K*, elastic bulk modulus; *G*, elastic shear modulus; *p*, density; *c*, cohesion; *φ*, internal angle of friction; *φ_f_*, dilation angle; and *σ_t_*, tension limit. Only *E*, *G*, and *ρ* need to be defined for the linear-elastic material. These parameters of the numerical modeling were presented in Table 6, where the data for the LWFC were derived from performance testing results in Section 2, and the other parameters were determined according to geological reports and the literature [34].

In addition, the foundation soils of clay layers were modeled as the modified cam-clay (MCC) model [34,35], which includes eight essential material parameters: current elastic bulk modulus (*K*), slope of the elastic swelling line (*κ*), slope of the normal consolidation line (*λ*), Poisson’s ratio (*v*), pre-consolidation pressure (*p_c_*_0_), stress ratio at the critical state (*M*), specific volume at the reference pressure on the normal consolidation line (*v_λ_*), and density of the model (*ρ*). The yield function of MCC corresponding to a particular consolidation pressure value *p_c_* has the following form:(11)$$f(p,q)=q^{2}+M^{2}p(p-p_{c})$$

The yield condition *f* = 0 is represented by an ellipse with horizontal axis *p_c_* and vertical axis *M* in the (*q*,*p*) plane (Figure 18c). The failure criterion is represented in the principal stress space by an ellipsoid of rotation about the mean stress axis (any section through the yield surface at a constant mean effective stress *p* is a circle). The potential function *g* corresponds to an associated flow rule:(12)g(p,q)=f(p,q)

To obtain the property parameters of clay at the site, undisturbed soil tests were conducted using the method in the literature [34,35]. The average values of soil properties for each layer are summarized in Table 7.

## 5. Numerical Results versus Field Measurements

### 5.1. Settlement

The measured and computed settlements of monitoring points S-1 and S-2 with elapsed stages are shown in Figure 19a,b, respectively. The numerical results closely matched the monitored data. Settlements increased during the LWFC filling stage and gradually stabilized due to ground consolidation at the intermission stage. A higher growth rate was observed at the A,B group filling stage, stabilizing 70 days later. Point S-2, near the centerline of the widened portion, showed a higher growth than S-1 throughout construction. Settlements at S-1 and S-2 were 5.5 mm and 7.2 mm after LWFC filling, and 16.8 mm and 19.2 mm after stabilization, respectively. LWFC filling caused nearly 71% of the total height but only 37.5% of the total settlement. It can be seen that using LWFC as the widened portion of high-speed railway, the post-construction settlement meets the requirement of no more than 30 mm.

### 5.2. Pressure

Measured and simulated pressures at P-1 to P-8 with elapsed stages are shown in Figure 20, with good curve-fitting for vertical pressure but poor for horizontal pressure. Different pressure behaviors were observed in the LWFC-filled embankment, with vertical pressure dropping 4–6 kPa at the bottom and 2–3 kPa at the middle during intermission, while horizontal pressure dropped 12–22 kPa. This phenomenon, confirmed by research in field tests, is that the pressure increased as LWFC filled, with a higher wet density at the casting stage, and then the dropping stage occurred as the LWFC’s weight decreased greatly upon drying. By LWFC filling completion, vertical pressures at the base and interlayer stabilized at 45 and 24 kPa, respectively, then at 76 and 45 kPa, which are below LWFC’s experimental material strength.

Measured and numerical horizontal deformations of monitored piles are shown in Figure 21, with good agreement between them. Deformation curves shifted away from the embankment’s centerline as filling height increased. Maximum deformations of NSP were 1.9 mm, 4.4 mm, and 6.4 mm after geotechnical cushion, LWFC, and A,B group filling, respectively, and 2.2 mm, 7.3 mm, and 10.7 mm for FSP. Other reinforced piles had similar deformations. The numerical modeling accurately captured the deformation and pressure trends in the LWFC-widened embankment compared to field data.

The numerical results were well consistent with the measured data, with low variations. The potential causes of variations between simulation and measurement were analyzed as follows: (1) model simplifications and assumptions, such as homogeneous material assumptions and idealized boundary conditions, may not fully represent real conditions; (2) uncertainty in material properties, due to variability and measurement errors, can cause significant differences between numerical predictions and actual behavior; (3) differences between assumed and actual construction practices, such as compaction levels and construction-induced disturbances, affect real-world outcomes; (4) the time-dependent nature of soil deformation and stress distribution, which may not be fully captured in simulations, leads to discrepancies.

In accordance with [36], the primary requirements for embankment filler and construction control are as follows: (1) the particle size of the fill below the subgrade should be less than 75 mm to enhance the overall performance of the embankment; (2) the settlement of the basement after construction should be less than 30 mm; and (3) the embankment fill should have a strength capable of withstanding 15 kPa and exhibit good water stability. According to the test results, LWFC exhibits excellent self-supporting and integrative properties. Specifically, LWFC with a density of 500 kg/m^3^ demonstrated a strength of 880 kPa, which was significantly higher than the standard requirement. Additionally, LWFC shows minimal performance degradation under the influence of water, thereby meeting the water stability requirements.

Based on the experimental data, it is preliminary determined that the performance of LWFC meets the requirements for use as a widened portion of high-speed railway.

Reference [37] researched the behavior of an 8-m (which was similar to the height in this study) widening embankment filled with conventional soil and found that the post-construction settlement of the embankment exceeded 50 mm and the vertical pressure at the base exceeded 110 kPa. However, the LWFC-widened embankment in this study had corresponding post-construction measurements of 19.2 mm and 76 kPa, respectively. This demonstrated that using LWFC instead of traditional fill materials can significantly reduce embankment deformation and stress.

## 6. Parametric Study

In accordance with the results from Chapters 5 and 6, the additional loading generated by the widened portion is mainly determined by the weight of the filler, and a lower self-weight will reduce the overall settlement deformation induced by the widened embankment. Meanwhile, the pile-soil structure beneath the widened portion effectively resists the load transmitted from above, thereby reducing the settlement as well. As can be seen, the vertical stiffness of the foundation is primarily dependent on the diameter and spacing of the piles. The newly constructed embankment in this study has a considerable height, and the retaining wall is a key element in maintaining the overall stability of the new portion. Therefore, to further investigate the behavior of LWFC-filled widened embankments adjacent to existing high-speed railway, the aforementioned four key factors were selected for the parametric study: the type of filler with different density, the thickness of the retaining wall, the diameter of the piles, and the offset of the piles. For each research variable, three typical parameter values were used.

### 6.1. Influence of Filler Type

The Ministry of Railways of China compiled the “Technical Guidelines for the Construction of Railway Subgrades for Passenger Dedicated Lines”, which includes the division standards of fillers used for different subgrade structural layers in 2005. The “Code for Design of Railway Subgrade” was promulgated in 2017 after long-term exploration and research and lists the types of fillers suitable for different layers of embankment structures, including the surface layer and the lower part, as well as the bottom of the embankment. Currently, the use of graded crushed stone is stipulated for the surface layer of the foundation bed, and the A and B group fillers or improved soil are stipulated for use below the foundation bed. In this study, an improved soil filler adopted in the literature [35] with a density of 1750 kg/m^3^ and A and B group fillers with the same properties as corresponding fillers in Table 5 were selected for analysis. The properties and parameters of the two types of fillers are presented in Table 8.

With reference to literature [38,39], the settlement of widened and existing embankments, as well as the horizontal deformation of retain walls, and piles, were captured to research the performance of widened embankments with the different fillers aforementioned.

Figure 22 shows the influence of the investigated fillers on the differential settlement under the bottom of the widened portion and the upper existing embankment. The larger amount and more significant differential settlement are shown in the widened and existing embankments when using traditional fillers. The maximum settlements of the widened embankment are 19.2 mm, 38.7 mm, and 41.5 mm for the LWFC, improved soil and A,B group filler, respectively. Correspondingly, the settlement deformations of the existing embankment are 4.2 mm, 7.9 mm, and 8.7 mm, respectively. The cause of this phenomenon is the lower density and excellent integrity of the LWFC filler compared to the contrast fillers with granular aggregate and higher compressibility (as pictured in Figure 22a). Hence, the less deformation and more optimized distribution of the pressure induced by self-weight are the typical characteristics of LWFC-widened embankment.

The horizontal deformation of the retain wall with height for different fillers is shown in Figure 23, where the maximum deformation at the toe of the wall are 2.1 mm, 2.4 mm, and 3.8 mm for LWFC filler, improved filler and A,B group filler, respectively. Correspondingly, 1.2 mm, 2.9 mm, and 4.2 mm at the head of the wall are computed for three kinds fillers. Furthermore, the maximum tensile stress on the outer surface of the wall are 0.07 MPa, 0.13 MPa, and 0.16 MPa when applying LWFC filler, improved soil, and A,B group filler. Generally, the LWFC filler, as a widened portion of the embankment, gives much smaller compressional stress on the retain wall during construction compared to the traditional fillers due to its lower density and excellent integrity, from which the thickness of the retain wall could be reduced to enhance economic performance.

Figure 24 presents the variation of the horizontal deformation of the piles with LWFC and the two traditional fillers. The trend of the horizontal deformation of the piles is similar, with a maximum value of 6.4 mm, 10.2 mm, and 11.3 mm moving toward the existing embankment at NSP, as well as 8.1 mm, 12.4 mm, and 13.9 mm moving outside of the existing embankment at FSP, respectively. Applying LWFC as widened filler within the embankment of a high-speed railway in this case decreases the maximum horizontal deformation of the NSP and FSP below the foundation by 37.2–43.4% and 34.7–41.7% based on the numerical modeling, respectively. Correspondingly, the stress of the piles is reduced by 39.5% to 53.1% on the NSP as well as 16.1% to 23.5% on the FSP, respectively.

Therefore, it can be seen that applying LWFC filler for widening embankment in high-speed railway effectively reduces the pressure on the foundation under the widened portion and retain wall, the differential settlement between the existing and widened embankments, as well as the horizontal deformation of the piles compared to the traditional soil for widening embankment.

### 6.2. Influence of Retaining Wall Thickness

It was noticed that the behavior of horizontal deformation of the retain wall based on numerical modeling for LWFC and the other two kinds of traditional filler is similar, with characteristics accompanied by a low value. Therefore, the thickness of the retain wall that could be reduced by computational support was investigated at 0.2 m, 0.5 m, and 0.8 m to research its influence on the lateral deformation of the LWFC-widened embankment in this case. For the retain wall with a lower thickness compared to the other parts of the widened portion, only the horizontal deformation of the wall is studied here. 

As expected, the curves (Figure 25) of horizontal deformation with height for different thicknesses were similar, with the toe of the wall moving outside of the widened portion and the head of the wall moving toward the embankment, and the deformation exhibited a positive correlation at the toe and a negative correlation at the head with thickness increasing, respectively. Increasing the wall thickness by 0.2 m to 0.8 m increased the maximum horizontal deformation to −0.8 mm at the toe and 0.9 mm at the head. The stress varied from 0.09 MPa to 0.06 MPa, with the thickness of the wall increasing from 0.2 m to 0.8 m. The results based on computation present a few variations of the maximum horizontal deformation and stress with a wall 0.2 m thick. 

### 6.3. Influence of Diameter and Offset of Piles

The diameter and offset, as the main influential factors affecting the stiffness of composite foundation, of the piles tried to minimize deformation resulted from the self-weight of the LWFC-filled widened embankment in this case were investigated. Moreover, the influence of the two factors on the settlement under the widened portion as well as the horizontal deformation of the NSP and the FSP were analyzed.

Figure 26 presents the variation of settlement under the widened embankment with different pile diameters and offsets. The settlement in the figure increases with a decrease in pile diameter and increases with an increase in pile offset, respectively. However, more significant variation in settlement and differential deformation between the pile block and soil adjacent caused by pile offset is observed compared to pile diameter. For example, the maximum settlement obtained by the pile of 0.5 m diameter located at the 1.5 m offset was lower than those induced by the same piles located at the 2.0 m and 2.5 m offsets by 42.2% and 80.9%, respectively. However, these percentages of increase in the maximum settlement decreased to 6.1% and 10.3% with an increase in pile diameter from 0.5 to 0.9 m by the 1.5 m offset. Figure 27 presents the relationship between the maximum settlement under the widened embankment and the pile diameter as well as the pile offset. It follows that optimizing the pile offset has a more effective influence on changing the settlement under the widened embankment compared to the pile diameter. 

The variations of horizontal deformation of the NSP and FSP with different pile offsets and diameters are shown in Figure 28a,b, respectively. A similar rule with settlement is that the horizontal deformation of piles along the depth increases with a decrease in pile diameter and increases with an increase in pile offset, respectively. As for NSP, the maximum deformation of the pile located at the 1.5 m offset decreased to 21.9% and 51.1% with an increase in pile diameter from 0.5 to 0.9, respectively. The decreased rate in the maximum horizontal deformation presented a range from 5.6% to 8.5% with an increase in pile diameter from 0.5 to 0.9 m by the 1.5 m offset, respectively. Also, a similar variation can be observed in the deformation of FSP, which was not presented here due to the page limit. This phenomenon could be attributed to the integral structural region created by the piles with a small offset, which makes a noteworthy increase in foundation stiffness to resisted to pressure induced by self-weight of LWFC widened portion, thus resulting in decreasing of the horizontal deformation under the widened embankment. 

## 7. Conclusions

In this study, the behavior of the widened embankment of a high-speed railway filled with LWFC with a design dry density of 550–650 kg/m^3^ was investigated via numerous performance tests, a field test, and numerical modeling. Moreover, a parametric study was further performed to examine the influence of four key factors on the deformation properties of the widened portion of the embankment. Based on the results of this study, the following conclusions can be drawn:

1. LWFC with a dry density of 550–650 kg/m^3^ has sufficient strength, stiffness, and impermeability to bear the loads and water immersion of a high-speed railway when it is applied as filler to the widened embankment. and all these properties exhibit a positive correlation with the dry density of LWFC. The amount of infiltration into the LWFC soaked in a low-pressure water environment can be neglected. However, the weakened strength of LWFC caused by water seepage is worthy of note.

2. A significant dropping stage of pressure inside the LWFC-filled widened embankment was observed during an intermittent period shortly after the filling of LWFC, and the excellent integral performance of LWFC formed with gradual solidification makes the dropping level of horizontal pressure more prominent compared to vertical pressure. The maximum vertical pressure and settlement inside the widened portion induced by the self-weight of LWFC, which account for nearly 71% of the total filling height, were only 59.2% and 37.5% of the corresponding eventual data after construction. 

3. According to the parametric study: (1) the density and the self-supporting properties of different backfills filled in the widened embankment of a high-speed railway remarkably affect the behavior of deformation inside the widened and existing portions of the embankment. Around two times of settlement in the widened and existing portions as well as more than 1.5 times of horizontal deformation inside piles under the widened embankment were captured when replacing LWFC filled as the widened portion with traditional filler; (2) The subtle variation of horizontal deformation and stress inside the retain wall were observed with the thickness of it changing from 0.2 m to 0.8 m. (3) The more significant variation in settlement of the foundation under the widened embankment and horizontal deformation of piles caused by pile offset is observed compared to pile diameter. 

4. The horizontal deformation of the retain wall can be negligible due to the LWFCs low density and excellent self-supporting properties. Thus, it indicates that making an appropriate reduction in the retain wall installed against the LWFC-widened embankment of high-speed railway will not generate the nonnegligible deformation with the centimeters of variation presented in the parametric study.

## 8. Recommendations

This work comprehensively utilized experimental and numerical methods to demonstrate that LWFC, as a lightweight and high-strength material, meets the requirements for application in high-speed railway embankment widening. However, there are still some issues that need further investigation and resolution for practical application and future work: (1) further exploration of lower-density LWFC is required to determine its suitability for embankment widening; (2) research can be conducted on actual construction practices to optimize LWFC construction techniques; and (3) optimization and identification of the optimal single filling height and length of LWFC in embankment widening should be pursued.

## Figures and Tables

**Figure 1 materials-17-04642-f001:**
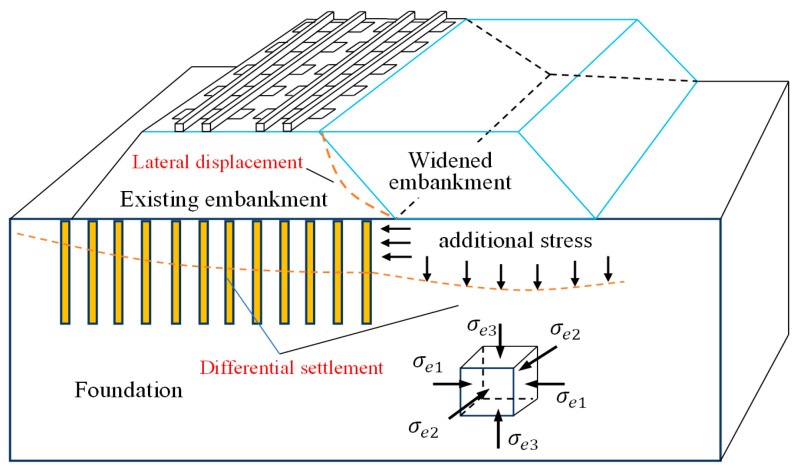
Distribution of deformation and internal forces in embankment of high-speed railway due to widening effects.

**Figure 2 materials-17-04642-f002:**
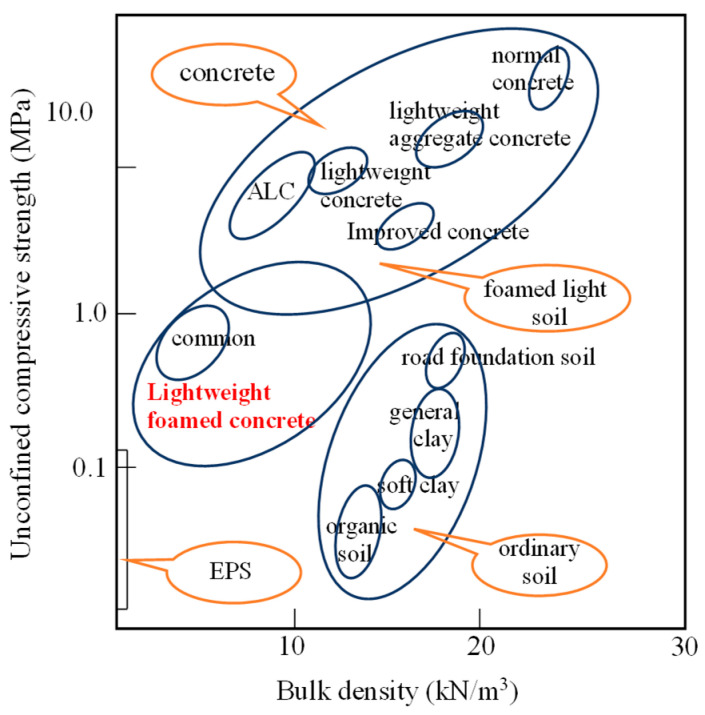
Density and unconfined compressive strength of different geotechnical fillers.

**Figure 3 materials-17-04642-f003:**
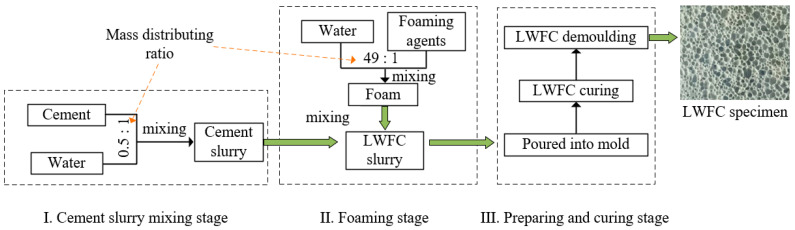
Preparing procedure of LWFC specimens.

**Figure 4 materials-17-04642-f004:**
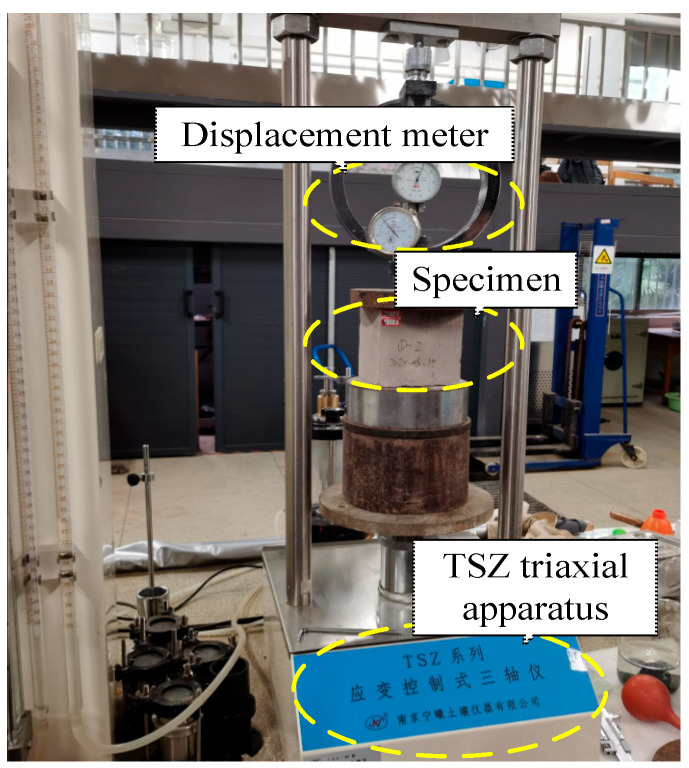
Compressive strength test.

**Figure 5 materials-17-04642-f005:**
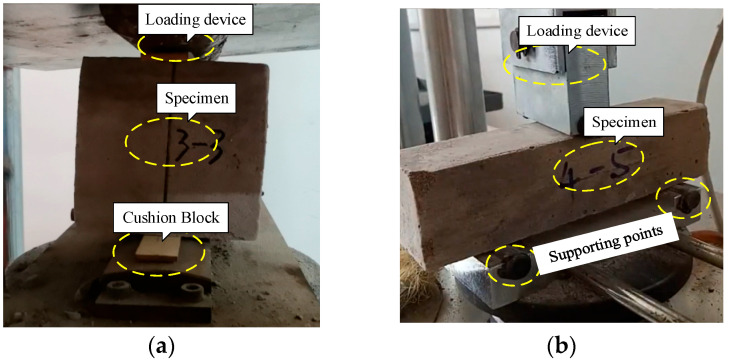
Tensile strength test. (**a**) Splitting test, (**b**) Bending test.

**Figure 6 materials-17-04642-f006:**
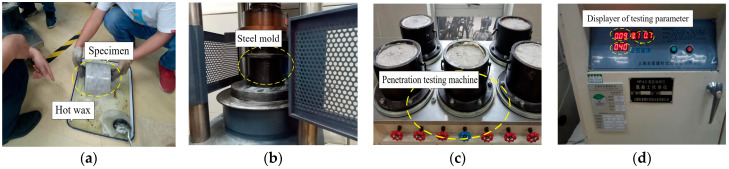
Permeability test. (**a**) Sealing wax on the side of specimen, (**b**) Forcing the specimen into mold, (**c**) Installing the specimen into penetrator, (**d**) Parameter setting before test.

**Figure 7 materials-17-04642-f007:**
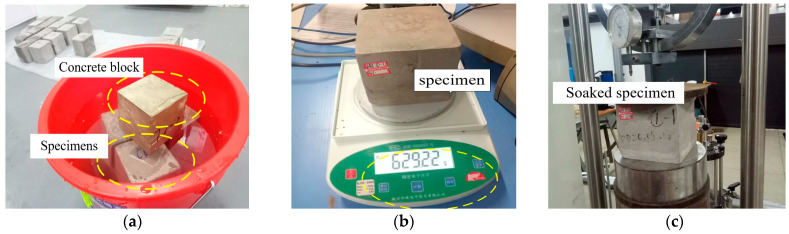
Compressive strength test of LWFC treated by water. (**a**) Immersing specimens into water, (**b**) Weighing the specimen, (**c**) Testing the soaked specimen.

**Figure 8 materials-17-04642-f008:**
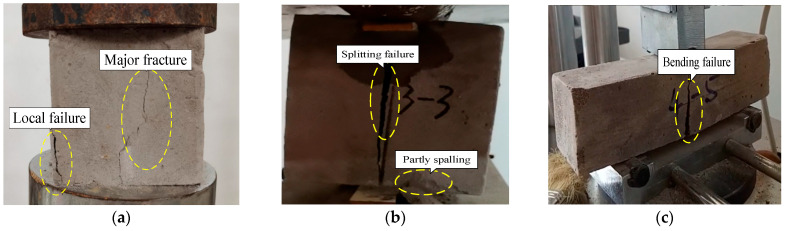
Failure diagram of specimens after compressive and tensile tests. (**a**) Compressive failure, (**b**) Splitting failure, (**c**) Bending failure.

**Figure 9 materials-17-04642-f009:**
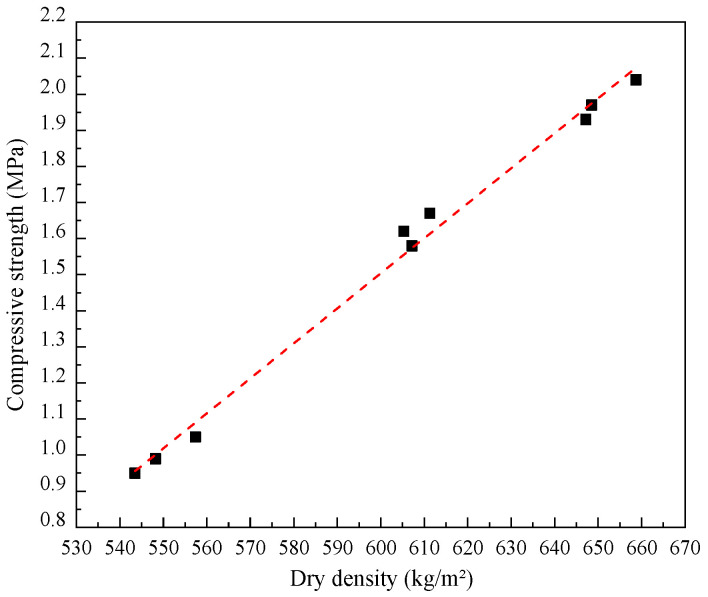
Fitted curve of compressive strength versus dry density.

**Figure 10 materials-17-04642-f010:**
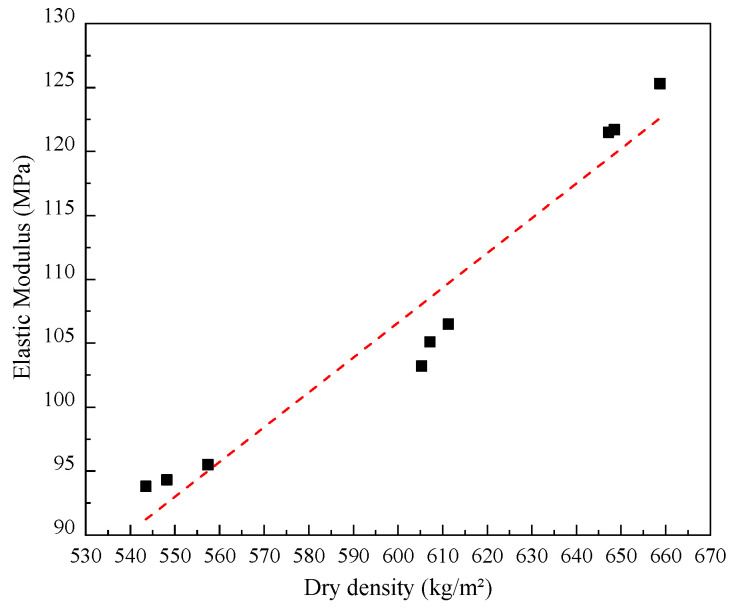
Fitted curve of elastic modulus versus dry density.

**Figure 11 materials-17-04642-f011:**
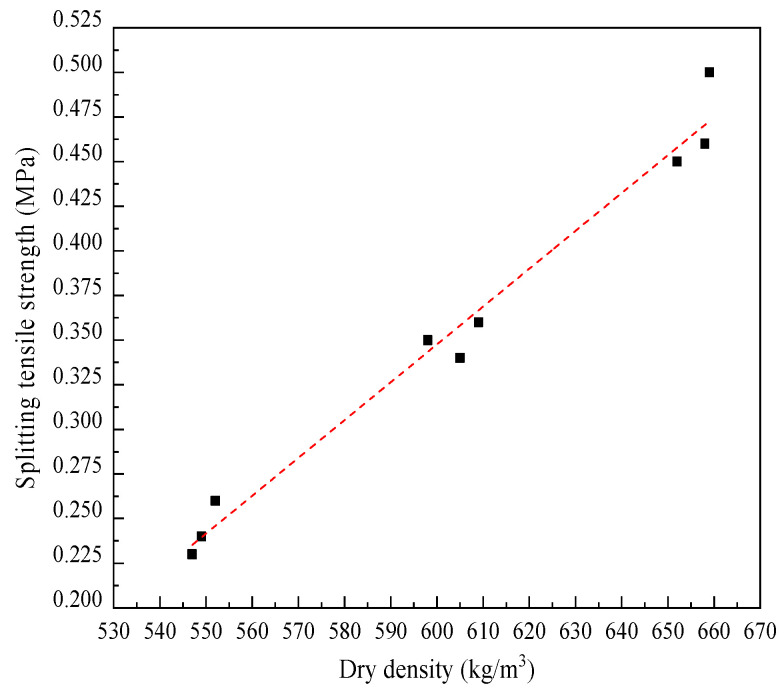
Fitted curve of splitting tensile strength versus dry density.

**Figure 12 materials-17-04642-f012:**
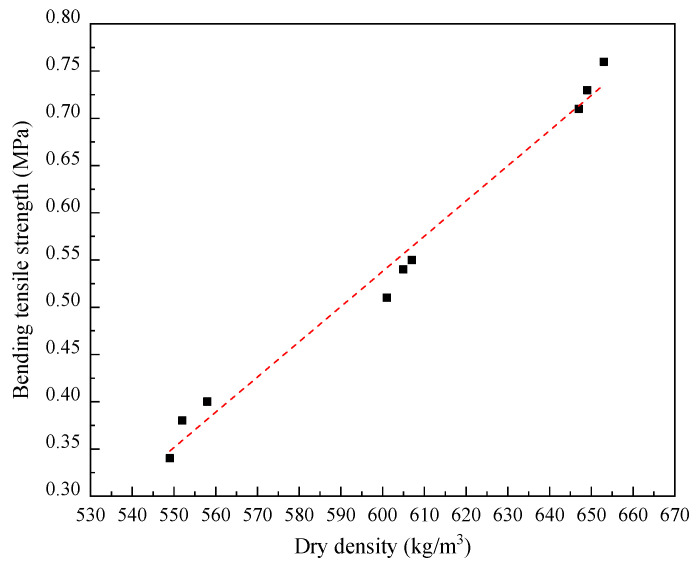
Fitted curve of bending tensile strength versus dry density.

**Figure 13 materials-17-04642-f013:**
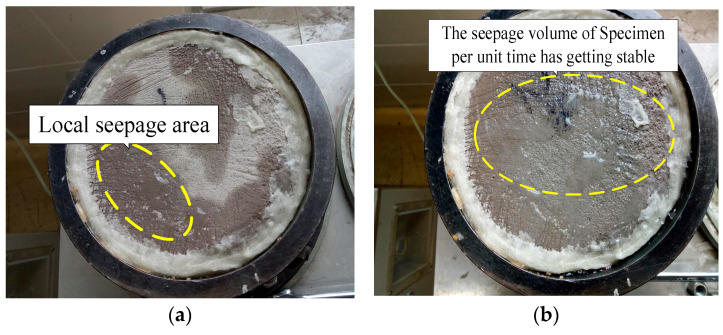
Permeability test. (**a**) Local osmotic phenomenon, (**b**) Seepage velocity reaches stable.

**Figure 14 materials-17-04642-f014:**
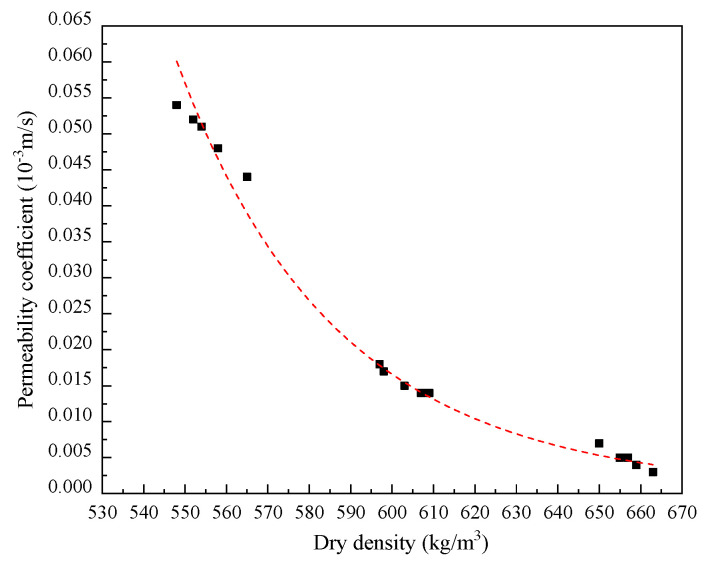
Fitted curve of permeability coefficient versus dry density.

**Figure 15 materials-17-04642-f015:**
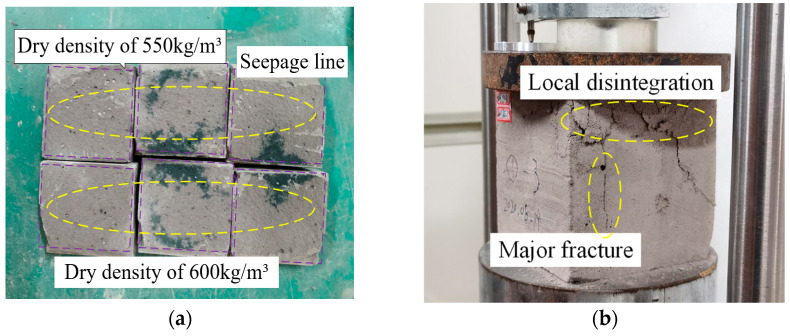
Specimens in the water immersion tests, (**a**) Cutting profile of the soaked specimens, (**b**) Compressed failure of the soaked specimens.

**Figure 16 materials-17-04642-f016:**
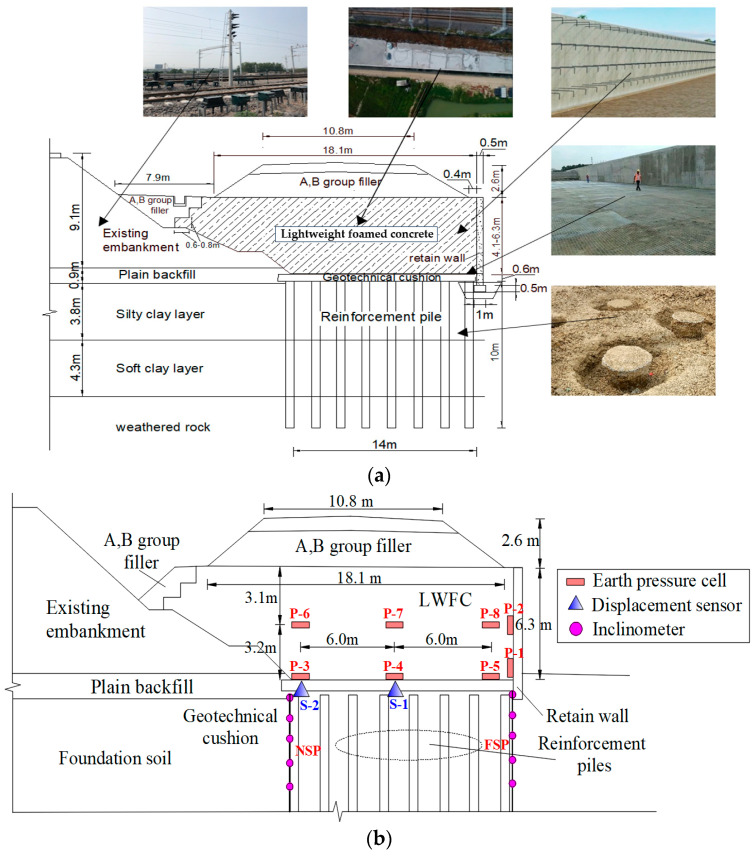
Field-test profile of the widened embankment. (**a**) Composition of the widened embankment, (**b**) Measuring-point layout in the cross-section.

**Figure 17 materials-17-04642-f017:**
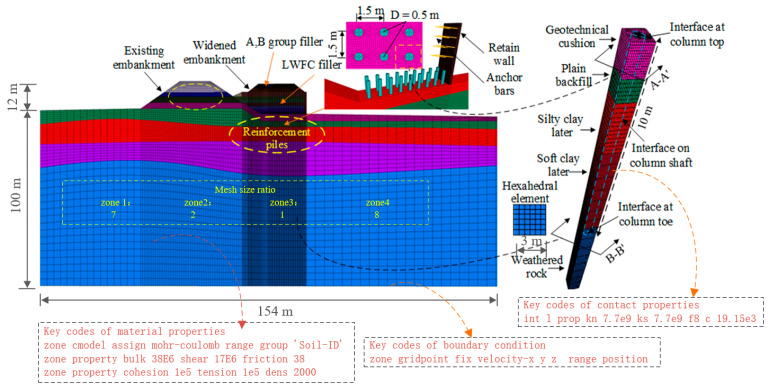
Numerical modeling for the analysis.

**Figure 18 materials-17-04642-f018:**
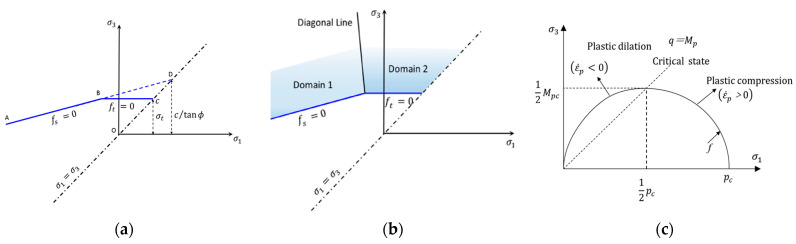
Strength criteria of different constitutive models, (**a**) Mohr–Coulomb failure criterion. (**b**) Mohr–Coulomb model used in the definition of the flow rule. (**c**) MCC failure criterion.

**Figure 19 materials-17-04642-f019:**
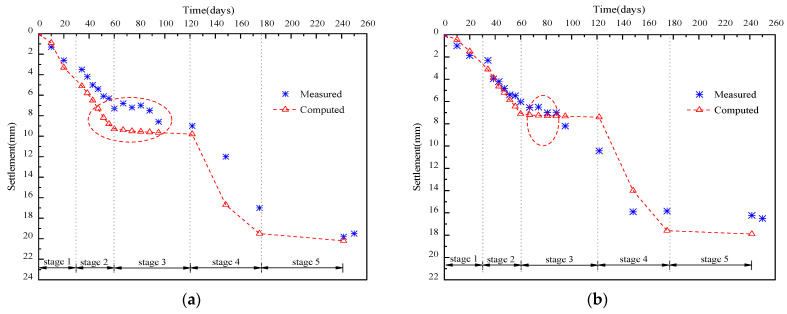
Settlement data comparison between measurement and simulation, (**a**) Settlement data comparison between measurement and simulation (S-1), (**b**) Settlement data comparison between measurement and simulation (S-2).

**Figure 20 materials-17-04642-f020:**
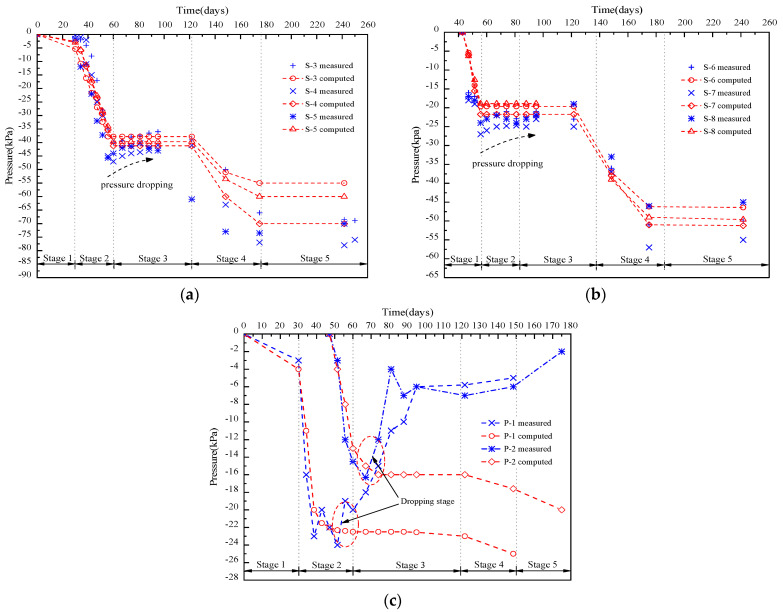
Pressure data comparison between measurement and simulation. (**a**) Monitoring points P-3,4,5. (**b**) Monitoring points P-6,7,8. (**c**) Monitoring points P-1,2.

**Figure 21 materials-17-04642-f021:**
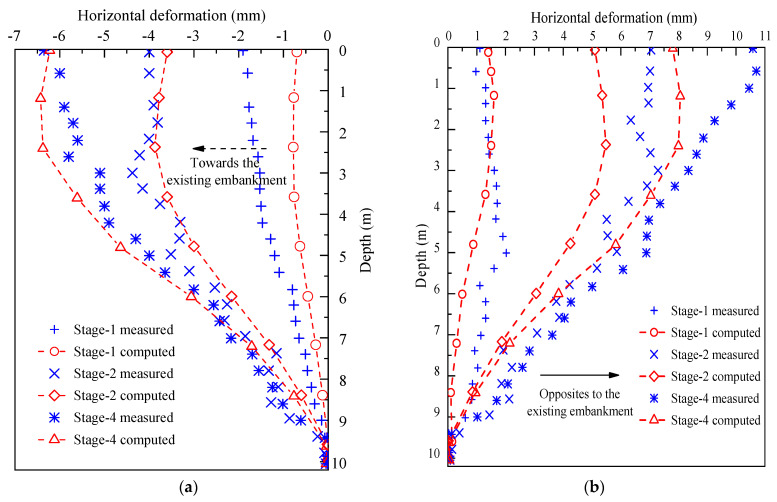
Horizontal deformation data comparison between measurement and simulation. (**a**) NSP. (**b**) FSP.

**Figure 22 materials-17-04642-f022:**
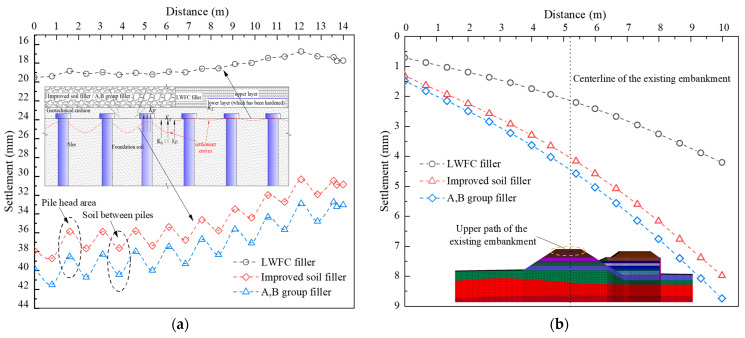
Influence of the filler type on settlement of embankments, (**a**) The widened embankment, (**b**) The existing embankment.

**Figure 23 materials-17-04642-f023:**
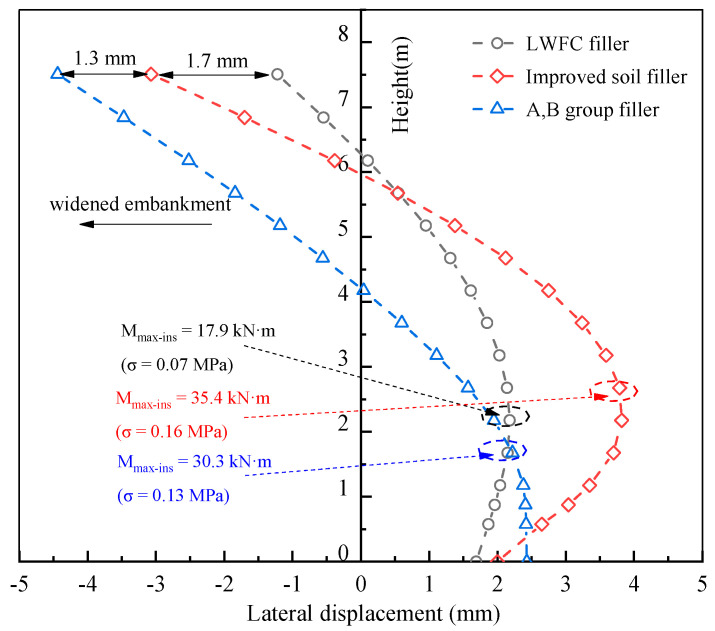
Influence of filler type on the horizontal deformation of the retaining wall.

**Figure 24 materials-17-04642-f024:**
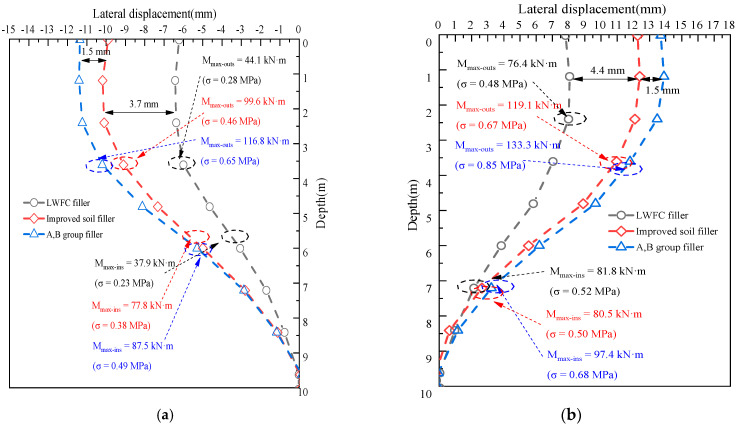
Influence of filler type on the horizontal deformation of piles. (**a**) NSP. (**b**) FSP.

**Figure 25 materials-17-04642-f025:**
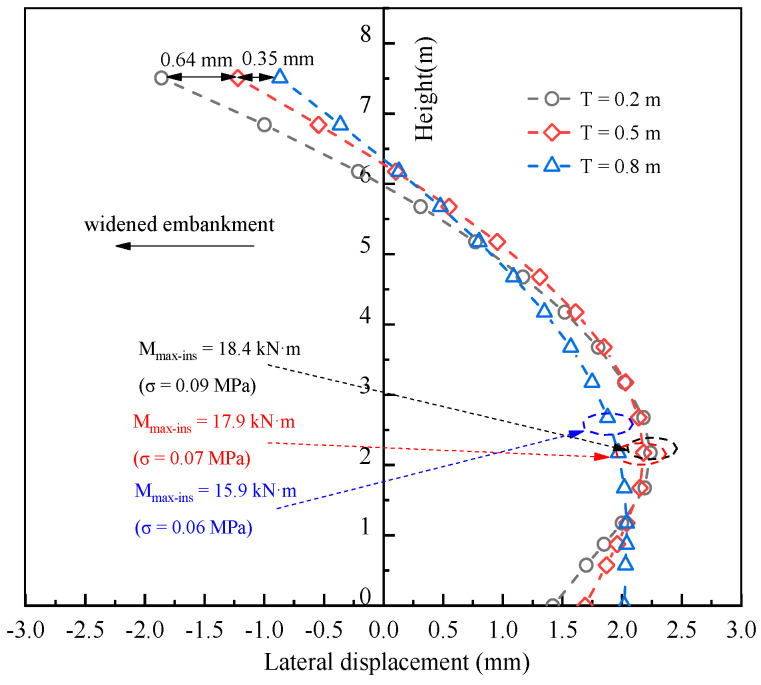
Influence of pile thickness on the horizontal deformation on the retaining wall.

**Figure 26 materials-17-04642-f026:**
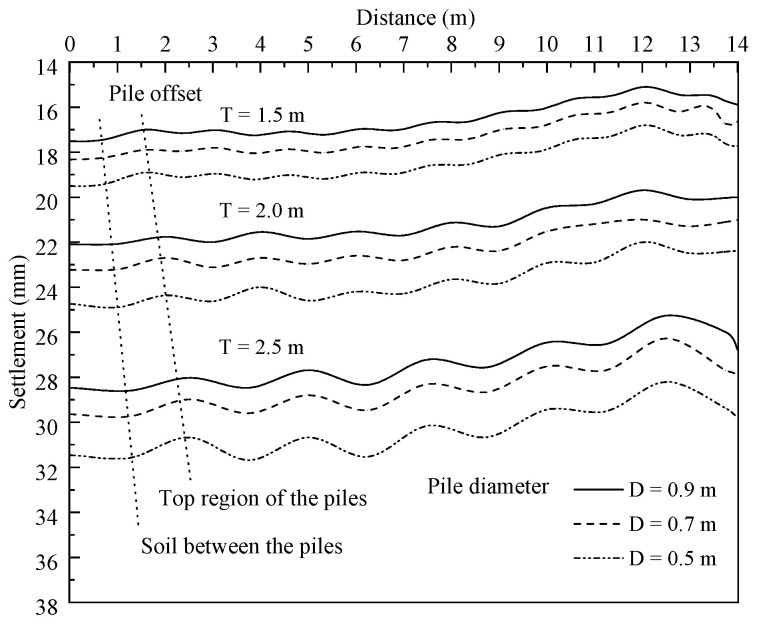
Influence of pile diameter and pile offset on settlement of foundation under the widened embankment.

**Figure 27 materials-17-04642-f027:**
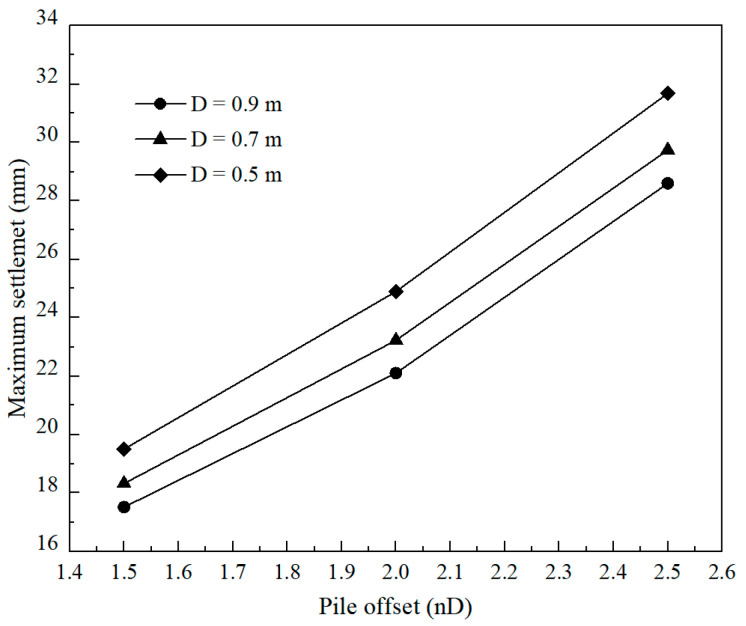
The maximum settlement of foundation under the widened embankment versus pile diameter and the offset.

**Figure 28 materials-17-04642-f028:**
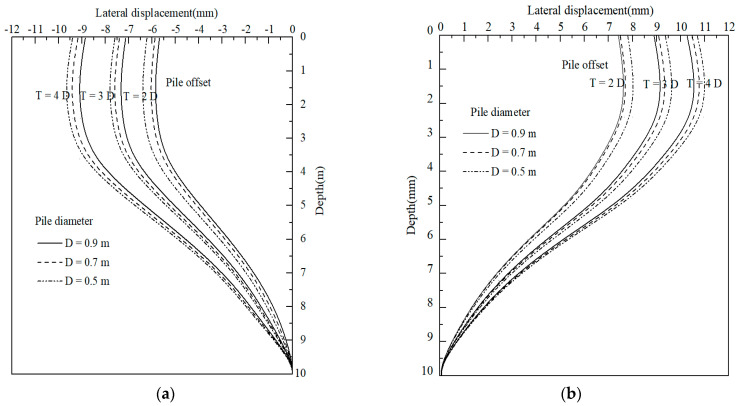
Influence of pile diameter and pile offset on the horizontal deformation of piles. (**a**) NSP. (**b**) FSP.

**Table 1 materials-17-04642-t001:** Physical properties of foam.

Type of Foam Agent	Compound Foam Agent
Foaming ratio Wet densities (kg/m Design dry density grades (kg/m^3^)	≥55
Foaming densities (kg/m^3^)	40–60
PH value	7.04
Foaming bleeding rate	≤20%
Increasing rate of wet density determined by defoaming test	≤10%

**Table 2 materials-17-04642-t002:** Mix proportions of LWFC with different design dry density.

Design Dry Density Grades (kg/m^3^)	Cement (kg)	Foam (m^3^)	Water (kg)
550	405	0.71	203
600	477	0.66	238
650	532	0.60	267

**Table 3 materials-17-04642-t003:** Results of performance tests.

Design Dry Density (kg/m^3^)	Compressive Strength (MPa)	Elastic Modulus (MPa)	Splitting Strength (MPa)	Bending Strength (MPa)	Permeability Coefficient (10^−3^ m/s)	Compressive Strength with the Effect of Water (MPa)
550	1.00	94.56	0.24	0.37	0.052	0.88
600	1.62	104.99	0.35	0.53	0.018	1.48
650	1.98	122.85	0.47	0.73	0.007	1.86

**Table 4 materials-17-04642-t004:** Construction process of the LWFC-filled embankment.

Stage of Construction Process	Construction Procedure at the Stage	Height of Filling (m)	Elapsed Time (Day)
1	Filling of geotechnical cushion	0.6	30
2	Filling of LWFC layer	6.3	60
3	Intermittent period	0.0	30
4	Filling of A,B group filler	2.6	30
5	Intermittent period	-	-

**Table 5 materials-17-04642-t005:** Physical properties of the site soil.

Soil Layer	Density (kg/m^3^)	Permeability Coefficient (×10^−4^)	Water Content (%)	Shear Strength (Direct Shear Testing)		Characteristic Value of Foundation Bearing Capacity (kPa)	Unconfined Pressive Strength (kPa)
				Cohesion (kPa)	Internal Friction Angle (°)		
Plain backfill	1920	1.16	15.2	5	7.0	/	/
Upper silty clay	1837	0.09	26.4	35	17.5	220	113
Lower soft clay	1949	0.41	35.8	30	14.5	180	110
Weathered rock	1980	0.58	19.8	300	31.0	600	/

**Table 6 materials-17-04642-t006:** Physical and mechanical properties of the backfill material, weathered rock, geotechnical cushion, piles, and retain wall.

Material		*E* (MPa)	*v* (MPa)	*ρ* (kg/m^3^)	*c* (kPa)	*φ* (°)	*φ^f^* (°)	*σ_t_* (MPa)
LWFC	Upper 0.8 m	117	0.23	640				
	Lower 5.5 m	98	0.23	580				
Piles		1500	0.21	2500				
Retain wall		20,000	0.21	2500				
Geotechnical cushion		45	0.3	2000	22	27	15	0.06
Existing embankment fill		290	0.23	2000	24	28	10	0
A, B group fill		290	0.23	2000	24	28	10	0
Weathered rock		19,000	0.22	1980	300	31	20	0.2

**Table 7 materials-17-04642-t007:** Physical and mechanical properties of the foundation clays.

Material	*K* (MPa)	*p_c_*_0_ (kPa)	*v*	*Ρ* (kg/m^3^)	*κ*	*M*	*λ*	*v_λ_*
Upper silty clay	18.3	191.6	0.3	1837	0.013	0.90	0.07	0.059
Lower soft clay	21.7	166.5	0.3	1949	0.010	0.86	0.07	0.067

**Table 8 materials-17-04642-t008:** Properties and parameters of fillers.

Material	*E* (MPa)	*v*	*ρ* (kg/m^3^)	*c* (kPa)	*φ* (°)	*φ^f^* (°)
Improved soil filler	70	0.25	1750	3.9	23	10
A, B group filler	290	0.23	2000	24	28	10

## Data Availability

The original contributions presented in the study are included in the article, further inquiries can be directed to the corresponding author.

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
