# Peer review of "Experimental and Numerical Study on Lightweight-Foamed-Concrete-Filled Widened Embankment of High-Speed Railway"

_materials, 2024, doi:10.3390/ma17184642_

Round 1
Reviewer 1 Report
Comments and Suggestions for Authors
This paper presents a comprehensive study on the performance of lightweight foamed concrete in widened embankments of high-speed railways. The authors conducted numerous experimental tests and numerical simulations to investigate the mechanical properties, water resistance, and deformation behavior of LWFC-filled embankments. The study provides valuable insights into the potential application of LWFC in railway construction. However, several aspects require further clarification and improvement before publication.
Introduction:
1. The paper lacks a clear and concise statement of the research gap and novelty. It is not entirely clear what differentiates this study from previous research on LWFC in embankment widening.
2. The methodology section could be improved by providing more details on the experimental setup, including specific equipment used, sample preparation procedures, and testing conditions.
Experimental:
3. The description of the LWFC preparation procedure is insufficient. The authors should provide more details on the mixing process, including the type of mixer used, mixing time, and temperature control.
4. For the LWFC specimen preparation, it may be useful to specify the water-cement ratio and provide justification for the selected value based on literature guidelines.
5. Additional details on the curing process for the LWFC specimens could help evaluate the testing conditions.
6. For the testing equipment and procedures, calibration information for the methods would strengthen the technical description.
7. Specifying the sample sizes for each test would support evaluation of the results.
8. The authors should clearly define the testing conditions for each experiment, including the temperature, humidity, and loading rate.
9. Discussion of quality control procedures during LWFC preparation could improve the reliability of the testing program.
10. The authors should discuss the statistical significance of the results, including the number of samples tested and the standard deviation of the data.
Field Test:
11. A brief soil investigation summary would provide useful geotechnical context for the test site.
12. Clarification on the embankment construction stages (e.g. time between stages) would be helpful.
Numerical:
13. Justification for the model size and boundary selection through references enhances the technical approach.
14. Description of the mesh generation technique supports evaluation of modeling accuracy.
15. Additional validation of the soil constitutive models could strengthen confidence in the numerical analysis.
Results and Discussion:
16. Direct comparison of key field vs. numerical data may better demonstrate modeling accuracy.
17. The discussion section needs to be strengthened by providing a more in-depth analysis of the results, comparing them with existing literature, and highlighting the implications for practical applications.
18. Potential causes of variations between measurement and simulation could be analyzed further.
Conclusions:
19. Links to specific results or findings could strengthen the conclusions presented.
20. Recommendations for practical application and future work would round out the technical contribution.
Author Response
Dear Reviewers
We sincerely wish you all the best. Thank you so much for taking the time out of your busy schedule to review my manuscript and providing many professional and valuable comments. We have made corresponding revisions to the manuscript based on your feedback. Below, we provide detailed responses to each of the issues you raised.
Introduction:
Comments 1: The paper lacks a clear and concise statement of the research gap and novelty. It is not entirely clear what differentiates this study from previous research on LWFC in embankment widening.
Response 1: Dear reviewer, we realized that we neglected to explain the innovation of this work and its differentiates from existing researches in the original manuscript. Based on your valuable comments, we have further supplemented the content reflecting the necessity of this work and its differences from previous studies in introduction, which have been marked with red color on page 2.
Comments 2: The methodology section could be improved by providing more details on the experimental setup, including specific equipment used, sample preparation procedures, and testing conditions.
Response 2: Dear reviewer, based on your precious comments. We have further added more contents of the experimental setup and testing conditions, which have been marked with red color on page 3 to 4.
Experimental:
Comments 3: The description of the LWFC preparation procedure is insufficient. The authors should provide more details on the mixing process, including the type of mixer used, mixing time, and temperature control.
Response 3: Dear reviewer, according to your request, we have added the contents of the LWFC preparation procedure with more detail, which have been marked with red color on page 3.
Comments 4: For the LWFC specimen preparation, it may be useful to specify the water-cement ratio and provide justification for the selected value based on literature guidelines.
Response 4: Dear reviewer, The mix ratio and preparation method of the LWFC adopted in this work were designed and implemented with reference to the “Ministry of Housing and Urban-Rural construction of the People's Republic of China. Foamed concrete: JG/T266—2011”in China. According to your comments, we have made supplementary explanations of this content in the manuscript.
Comments 5: Additional details on the curing process for the LWFC specimens could help evaluate the testing conditions.
Response 5: Dear reviewer, the curing method and process for the LWFC specimens in this work was completed with reference to the “Ministry of Housing and Urban-Rural construction of the People's Republic of China. Foamed concrete: JG/T266—2011” as well. In the stage of writing the original manuscript, we did not introduce the curing process in detail due to the conventional methods were adopted. In accordance with your comments, we have supplement some contents for additional details on the curing process of LWFC specimens in this work.
Comments 6: For the testing equipment and procedures, calibration information for the methods would strengthen the technical description.
Response 6: Dear reviewer, based on your comments. We have supplemented the calibration information for the testing equipment and procedures, which have been marked with red color on page 4.,
Comments 7: Specifying the sample sizes for each test would support evaluation of the results.
Response 7: Dear Reviewer, in this work, the sizes of specimens in one type of test were identical. The shape and size of the tested specimens used in different kinds of tests mainly refer to the code requirements [33]. We have checked this content in the experimental introduction
Comments 8: The authors should clearly define the testing conditions for each experiment, including the temperature, humidity, and loading rate.
Response 8: Dear Reviewer, based on your precious comments, we have clearly defined the contents of temperature, humidity, and loading rate in the tests, which have been marked with red color on page 4.
Comments 9: Discussion of quality control procedures during LWFC preparation could improve the reliability of the testing program.
Response 9: Dear reviewer, based on your precious comments, we have supplemented the discussion of quality control procedures during LWFC preparation, which have been marked with red color on page 4.
Comments 10: The authors should discuss the statistical significance of the results, including the number of samples tested and the standard deviation of the data.
Response 10: Dear Reviewer, In the performance tests, we conducted three samples in each type of test according to the standards and used the average value as the representative results. From a statistical perspective, these data may be limited, and analyzing their statistical parameters may lack persuasiveness as well. Therefore, we kindly hope you allow us to retain the original description. In future studies, we will fully consider your suggestions and conduct more extensive data analysis.
Field Test:
Comments 11: A brief soil investigation summary would provide useful geotechnical context for the test site.
Response 11: Dear reviewer, based on your precious comments, we have supplemented the soil investigation summary to improve the context for the test site, which have been marked with red color on page 9.
Comments 12: Clarification on the embankment construction stages (e.g. time between stages) would be helpful.
Response 12: Dear reviewer, during the field testing, we carried out measurements referring to the filling height of LWFC embankment, and we can guarantee to provide accurate test data corresponding to the height. Therefore, we kindly hope that you allow us to retain this content.
Numerical:
Comments 13: Justification for the model size and boundary selection through references enhances the technical approach.
Response 13: Dear reviewer, based on your precious comments, the required justification for the model size and boundary selection through references has been supplemented. In addition, the information of dimensions of the modelled area, key codes used and geometrical ratio of the meshes in Figure 17 has been supplemented, which have been marked with red color on page 10.
Comments 14: Description of the mesh generation technique supports evaluation of modeling accuracy.
Response 14: Dear reviewer, based on your precious comments, the description for the mesh generation technique has been added in the manuscript, which have been marked with red color on page 10.
Comments 15: Additional validation of the soil constitutive models could strengthen confidence in the numerical analysis.
Response 15: Dear reviewer, thank you for your professional and perceptive comments. In this work, we refer to the existing literature to determine the constitutive models of the FE model. Due to the restrictions of the research topic and length, we did not make a comparative study for researching the comparisons of different constitutive models. we kindly hope that you allow us to retain this content.
Results and Discussion:
Comments 16: Direct comparison of key field vs. numerical data may better demonstrate modeling accuracy.
Response 16: Dear reviewer, thank you for your valuable comments, and we have improved the content with a direct comparison of field measurement with numerical data to make it better for demonstrating modeling accuracy,which have been marked with red color on page 13 to 14.
Comments 17: The discussion section needs to be strengthened by providing a more in-depth analysis of the results, comparing them with existing literature, and highlighting the implications for practical applications.
Response 17: Dear reviewer, based on your valuable comments, we have supplemented the discussion section for comparing the results in this work with existing literature, which have been marked with red color on page 15.
Comments 18: Potential causes of variations between measurement and simulation could be analyzed further.
Response 18: Dear reviewer, based on your comments, the potential causes of variations between measurement and simulation could have been analyzed in the manuscript by us to further improved the content, which have been marked with red color on page 14.
Conclusions:
Comments 19: Links to specific results or findings could strengthen the conclusions presented.
Response 19: Dear reviewer, thank you for your valuable comments. In this work, we studied the effect of LWFC in the embankment widening of high-speed railway via laboratory test and field test. The selective data were linked to the conclusions presented. we kindly hope that you allow us to retain this content.
Comments 20: Recommendations for practical application and future work would round out the technical contribution.
Response 20: Dear reviewer, we all appreciate your invaluable comments. And we have added recommendations for practical application and future work in the manuscript, which have been marked with red color on page 20.
Finally, we would like to once again thank you for your professional guidance on our manuscript. If there are any issues with the revised version, we kindly ask for your valuable feedback, and we will make every effort to revise it until it meets your requirements. We wish you good health and success in your work.

Reviewer 2 Report
Comments and Suggestions for Authors
The performance of lightweight foamed concrete in widened embankments of high-speed railway is studied. Experimental and numerical work was used in this study.
The following questions need to be addressed:
1) What is the rationale for a design density of 550-650 kg/m³?
2) What is the significance of four factors in the parametric study? Discuss in details.
Comments on the Quality of English LanguageThe performance of lightweight foamed concrete in widened embankments of high-speed railway is studied. Experimental and numerical work was used in this study.
The following questions need to be addressed:
1) What is the rationale for a design density of 550-650 kg/m³?
2) What is the significance of four factors in the parametric study? Discuss in details.
Author Response
Dear Reviewers
We sincerely wish you all the best. Thank you so much for taking the time out of your busy schedule to review my manuscript and providing many professional and valuable comments. We have made corresponding revisions to the manuscript based on your feedback. Below, we provide detailed responses to each of the issues you raised.
Comments 1: What is the rationale for a design density of 550-650 kg/m³?
Response 1: Dear reviewers, thank you for your valuable comments. In this work, LWFC with a density of 550-650 kg/m³ is selected for the following reasons: In the revised manuscript [25], scholar Othman has conducted a study on the relation between density and strength of LWFC. We preliminarily concluded from Othman’ researches that LWFC with a density of 500kg/m³ may meet the requirements of embankment widening in the high-speed railway. Therefore, we decided to take the LWFC with a density of 550-650 kg/m³ as the research object and adopted it for verification in the field test.
Comments 2: What is the significance of four factors in the parametric study? Discuss in details.
Response 2: Dear Reviewer, the original manuscript did not provide an explanation for why we studied the four factors. According to your constructive comment, We have included a detailed explanation of the reasons and significance for selecting these factors for our study in Chapter 6.
Finally, we would like to once again thank you for your professional guidance on our manuscript. If there are any issues with the revised version, we kindly ask for your valuable feedback, and we will make every effort to revise it until it meets your requirements. We wish you good health and success in your work.

Reviewer 3 Report
Comments and Suggestions for Authors
The publication focuses on examining the performance of lightweight foam concrete (LWFC) in widened embankments of high-speed railway lines. The study includes tests of strength, permeability, and water immersion of LWFC with different densities. The results show that LWFC meets the requirements for bearing capacity and impermeability, although its strength is reduced by water. Field tests and numerical modeling analyze the behavior of deformations and pressure in filled LWFC sections. Parametric study identifies the influence of four factors on the performance of LWFC in widened embankments, suggesting that reducing the thickness of the retaining wall and optimizing pile displacement can reduce settlement and deformation, which is important in designing stable railway embankments.
To improve the quality of the work, the reviewer suggests paying attention to the following issues:
1. The introduction discusses many studies and methods used to improve railway infrastructure but does not present a specific gap in the literature being addressed in this work. This lack of information makes it difficult to understand why this work is needed.
2. The introduction makes many references to past studies, indicating a solid knowledge of the research state in this field. However, the reviewer notices gaps in references to the latest research, suggesting that the most up-to-date information may not have been considered.
3. The reviewer finds that the description of the research procedure is detailed and allows for a thorough understanding of the testing methods. However, they believe it may be too technical for readers unfamiliar with the subject, making it difficult for them to understand the essence of the research.
4. The text does not reference previous studies on LWFC performance tests. Could the authors present how testing methods have been adapted or improved?
5. The manuscript lacks an explanation of why specific studies were chosen and what expected benefits result from conducting these tests. It would be worth discussing which features of LWFC are crucial for its effective use in widening railway embankments.
6. The text lacks a comparison of the obtained results with industry standards. Such a comparison could provide a better understanding of the effectiveness of LWFC in the context of existing materials and methods.
7. The results of compressive and tensile strength tests were obtained based on laboratory samples, which may not accurately reflect the conditions of real embankments. This fact limits the practical application of the obtained results.
8. The text presents test results but lacks a deeper interpretation of how these results affect the potential application of LWFC in widening railway embankments. It would be worth discussing whether the obtained results meet technical requirements and what steps can be taken to improve the material's performance.
The conducted study contributes significantly to understanding the behavior of widened embankments in the context of high-speed railways. The use of various research methods has led to comprehensive and versatile results that may be useful for designers and engineers working on similar projects.
Author Response
Dear Reviewers
We sincerely wish you all the best. Thank you so much for taking the time out of your busy schedule to review my manuscript and providing many professional and valuable comments. We have made corresponding revisions to the manuscript based on your feedback. Below, we provide detailed responses to each of the issues you raised.
Comments 1: The introduction discusses many studies and methods used to improve railway infrastructure but does not present a specific gap in the literature being addressed in this work. This lack of information makes it difficult to understand why this work is needed.
Response 1: Based on the valuable opinions of reviewer , we have added the content in the introduction to show why this work is needed.
Comments 2: The introduction makes many references to past studies, indicating a solid knowledge of the research state in this field. However, the reviewer notices gaps in references to the latest research, suggesting that the most up-to-date information may not have been considered.
Response 2: Dear reviewer, according to your comments, we have reviewed and studied recent literature. However, we still have not found any relevant studies on the application of LWFC in high-speed railway embankment widening. In our future research, we will ensure a thorough investigation of the latest research findings.
Comments 3: The reviewer finds that the description of the research procedure is detailed and allows for a thorough understanding of the testing methods. However, they believe it may be too technical for readers unfamiliar with the subject, making it difficult for them to understand the essence of the research.
Response 3: Dear reviewer, we sincerely thank you for the valuable comments. We try our best to elaborate the research methods and results of each part in the manuscript and we trust the you can find from the length of the manuscript. If there is any ambiguity, we will try to improve this problem to help readers to understand the specific process of this research.
Comments 4: The text does not reference previous studies on LWFC performance tests. Could the authors present how testing methods have been adapted or improved?
Response 4: Dear reviewer. in this work, in order to obtain material properties more accurately, we added adscititious displacement meter in different strength tests to measure the displacement of tested specimen under different load levels. And provide more real data for the subsequent field test. This has rarely been used in previous literature studies. In addition, we have improved the description in the experimental section of the manuscript.
Comments 5. The manuscript lacks an explanation of why specific studies were chosen and what expected benefits result from conducting these tests. It would be worth discussing which features of LWFC are crucial for its effective use in widening railway embankments.
Response 5: Dear reviewer, in this work, we focused on investigating the strength and durability of LWFC fillers. The material performance test and the field test were performed respectively to validate the effectiveness of using LWFC as the widened embankment in high-speed railway. According to your valuable comment, we have added content to the original manuscript to explain why specific studies were chosen and what expected benefits result from conducting these tests in introduction.
Comments 6: The text lacks a comparison of the obtained results with industry standards. Such a comparison could provide a better understanding of the effectiveness of LWFC in the context of existing materials and methods.
Response 6: Dear reviewer, your comment has made us realize that we have overlooked this important point. Now we have added comparison of the obtained results with industry standards to the original article, which has been marked on page 14.
Comments 7: The results of compressive and tensile strength tests were obtained based on laboratory samples, which may not accurately reflect the conditions of real embankments. This fact limits the practical application of the obtained results.
Response 7: Dear reviewer, in order to fully study the performance behavior of LWFC filled in the widening embankment of the high-speed railway, all the specimens used for compressive and tensile strength tests were taken from the filler in the field test.
Comments 8: The text presents test results but lacks a deeper interpretation of how these results affect the potential application of LWFC in widening railway embankments. It would be worth discussing whether the obtained results meet technical requirements and what steps can be taken to improve the material's performance.
Response 8: Dear Reviewer, this work primarily investigated the performance of LWFC material for high-speed railway embankment widening. The presented results were used to assess whether it meets the requirements for high-speed railway embankment fill. This aspect was not detailed in the original manuscript. Therefore, in light of your suggestions, we have refined and supplemented the explanation including a deeper interpretation of results in this work ,and comparison of which with those of the existing literature using different embankment widened filler.
Finally, we would like to once again thank you for your professional guidance on our manuscript. If there are any issues with the revised version, we kindly ask for your valuable feedback, and we will make every effort to revise it until it meets your requirements. We wish you good health and success in your work.

Reviewer 4 Report
Comments and Suggestions for Authors
Dear Authors,
This research paper has joined experimental aspects, numerical modelling and practical engineering application. Structure and content are valuable.
Some points to refine for graphical parts and for references.
Thanks and best wishes
1)Arrange columns width of table 3.
2) In fig. 4 mention the type of transducers for strain measurement.
3)More comments in caption of fig. 13 for the specimen description.
4) Fig 16 a and 16 b seem to be too small.
5) In caption of fig. 17, insert the dimensions of the modelled area, code used and geometrical ratio of the meshes of embankment and subsoil at contact.
6)Improve details of caption of fig.18.
7) Background and references. In this construction field, researches and application have shown a great spreading in the last years, with the use of foams, light materials, geosynthetics, recycled materials and so on. Literature review has been appreciated, only two more recent papers are suggested to enlarge backgroiund for these topics.
(a) Use of recycled and secondary aggregates in foamed concretes. (2012) By Jones, R. et al. in Magazine of Concrete Res., 64, 513–525. doi.org/10.1680/macr.11.00026
(b) Analysis of the Importance of the Filling Material Characteristics Injected Around the Precast Concrete Lining in the Microtunnelling Technology. (2023) by Benato A. et al. in Geotechnical and Geological Engineering, Open Access, Vol. 41, Issue 5, Pag. 2775 – 2785, doi 10.1007/s10706-023-02405-9
Author Response
Dear Reviewers
We sincerely wish you all the best. Thank you so much for taking the time out of your busy schedule to review my manuscript and providing many professional and valuable comments. We have made corresponding revisions to the manuscript based on your feedback. Below, we provide detailed responses to each of the issues you raised.
Comments 1: Arrange columns width of table 3.
Response 1: Dear reviewer, in accordance with your comments, We have adjusted the width of the columns in Table 3.
Comments 2: In fig. 4 mention the type of transducers for strain measurement.
Response 2: Dear reviewer, in the strength performance test, the adscititious displacement meter was used to test the deformation data, according to which the strain data was also obtained. We are sorry that this was not explained in the original manuscript. According to your valuable comments, we have made modifications and improvements to this part.
Comments 3: More comments in caption of fig. 13 for the specimen description.
Response 3: According to the expert review, we have analyzed and explained Figure 13 with more content, in which we provide a detailed account of the permeability tests conducted on LWFC specimens. It discusses the osmotic phenomenon observed during the tests, the detection of early water movement, the progression to a steady-state flow condition. Additionally, it highlights the variability in permeability due to the heterogeneous distribution of bubbles within the material.
Comments 4: Fig 16 a and 16 b seem to be too small.
Response 4: According to the reviewer's valuable comments, the size of the original draft figure 16 has been adjusted and enlarged.
Comments 5: In caption of fig. 17, insert the dimensions of the modelled area, code used and geometrical ratio of the meshes of embankment and subsoil at contact.
Response 5: According to the opinions of the reviewer, the information of dimensions of the modelled area, key codes used and geometrical ratio of the meshes in Figure 17 has been supplemented.
Comments 6: Improve details of caption of fig.18.
Response 6: Dear reviewer, the Strength criteria of different constitutive models presented in FIG. 18 are mainly derived and modified from the reference of soil mechanics. We kindly hope you allow us to retain the original details.
Comments 7: Background and references. In this construction field, researches and application have shown a great spreading in the last years, with the use of foams, light materials, geosynthetics, recycled materials and so on. Literature review has been appreciated, only two more recent papers are suggested to enlarge backgroiund for these topics.
(a) Use of recycled and secondary aggregates in foamed concretes. (2012) By Jones, R. et al. in Magazine of Concrete Res., 64, 513–525. doi.org/10.1680/macr.11.00026
(b) Analysis of the Importance of the Filling Material Characteristics Injected Around the Precast Concrete Lining in the Microtunnelling Technology. (2023) by Benato A. et al. in Geotechnical and Geological Engineering, Open Access, Vol. 41, Issue 5, Pag. 2775 – 2785, doi 10.1007/s10706-023-02405-9
Response 7: According to the reviewer's valuable comments, We have perused the literatures "Use of recycled and secondary aggregates in foamed concretes" and "Analysis of the Importance of the Filling. Material Characteristics Injected Around the Precast Concrete Lining in the Microtunnelling Technology ", the two references in related fields have been cited to enrich the content of this work and marked with red color.
Finally, we would like to once again thank you for your professional guidance on our manuscript. If there are any issues with the revised version, we kindly ask for your valuable feedback, and we will make every effort to revise it until it meets your requirements. We wish you good health and success in your work.

Round 2
Reviewer 1 Report
Comments and Suggestions for Authors
I appreciate the authors' effort in addressing the issues I raised. The revisions and clarifications provided have significantly improved the manuscript.